# Prevalence and factors associated with generalized anxiety disorder among patients with chronic pain: A single center cross sectional study in Malaysia

**Norlaila Abd Rahman**[1], **Suthahar Ariaratnam**[1]*, **Nurul Azreen Hashim**[1], **Zahir Izuan Azhar**[2]

**1** Department of Psychiatry, Faculty of Medicine, Universiti Teknologi MARA (UiTM), Sungai Buloh, Selangor, Malaysia, **2** Department of Public Health Medicine, Faculty of Medicine, Universiti Teknologi MARA (UiTM), Sungai Buloh, Selangor, Malaysia

* suthaharariaratnam@yahoo.com.au

## Abstract

### Background

Managing chronic pain was not only a major challenge but also a source of significant disability associated with mental illness. Studies on generalized anxiety disorder (GAD) in chronic pain population was rather limited. This study was aimed to determine the prevalence of GAD and its associated factors among patients attending a pain clinic at a general hospital.

### Methods

This cross-sectional study recruited 201 patients. The Hospital Anxiety Depression Scale (HADS) was used to determine anxiety level. Subsequently, patients who had scored 8 and above on the HADS were interviewed using Mini International Neuropsychiatric Interview (M.I.N.I) to ascertain the diagnosis of GAD. Whilst the Numerical Rating Scale (NRS) assessed pain severity. Multiple logistic regression analysis was used to determine factors associated with GAD.

### Results

Among those patients with chronic pain, the prevalence of GAD was 18.9%. Gender (AOR:7.94; 95% CI:2.34, 26.93), duration of the pain (AOR:1.30; 95% CI:1.03,1.63) and pain severity (AOR:18.75; CI:1.23,285.13) were significant factors associated with GAD.

### Conclusion

GAD is a prevalent condition among chronic pain patients.

**Data Availability Statement:** Our study had involved human research participant data. Hence, there are ethical and legal restrication on sharing of de-identified or anonymized data because these

information contain potentially identifying or sensitive patient information. However,data requests may be sent to the following ethics committees 1. Chairperson, Medical Research & Ethics Committee, Ministry of Health Malaysia, Kompleks Institut Kesihatan Negara, Blok A, No 1, Jalan setia Murni U 13/52, Seksyen U 13, Bandar Setia Alam, 40170 Shah Alam, Selangor, Malaysia. Tel: +603-33628888 Email: mrecsec@moh.gov.my 2. Chair, Universiti Teknologi MARA Research Ethics Committee, Aras 3, Bangunan Wawasan, 40450 Shah Alam, Selangor, Malaysia. Tel: +603-55442004 Fax: +603-55442070.

**Funding:** The author(s) received no specific funding for this work.

**Competing interests:** The authors have declared that no competing interests exist.

## Introduction

The International Association for the Study of Pain (IASP) had defined pain as "An unpleasant sensory and emotional experience associated with actual or potential tissue damage or described in terms of such damage" [1]. It is one of the features for many disease diagnoses which could determine the severity and activity of an underlying condition. Chronic pain is the pain that persists for three months or longer and currently troubles the patient either all of the time or as an on and off phenomenon [1].

Anxiety disorders were a group of conditions characterized by feelings of nervousness, excessive fear, panic and anticipation of future threats associated with clinically significant impairment of social and occupational functioning [2]. Some people used anxiety as a form of coping mechanism to lower somatic arousal, which was linked to pain, making them more vulnerable to develop generalized anxiety disorder (GAD). GAD is defined as a persistent anxiety over a variety of events and activities, such as job concerns, routine chores, relationships, and health issues [3].

Generally, various research had elucidated that anxiety and depression were commonly encountered in patients with pain [4–6]. These studies employed screening tools such as Patient Health Questionnaire –8 (PHQ-8) and Generalised anxiety disorder-7 (GAD-7) to determine presence of anxiety and/or depression which were essentially non diagnostic entities.

Furthermore, although de Heer et al. [7] utilized Composite International Diagnostic Interview (CIDI) which was as a diagnostic tool, the ensuing diagnostic variables were subcategorized as anxiety and depressive disorders among pain participants.

Additionally, most studies relied on self-report measures of non-specific psychological distress [8] and had not particularly explored GAD. The use of a structured diagnostic interview would improve the detection of GAD as well as provide a better understanding of GAD and its associated factors among patients with chronic pain.

Specifically, a recent review performed among chronic pain patients found that there were significant factors associated with depression and/or anxiety, namely intensity of chronic pain, severity of cancer related pain, daily persistent headache, long working hours, and monotonous work. However, the area of search for psychiatric diagnoses in that review was confined to depressive and/or anxiety disorders [9].

While a large study involving chronic low back pain among Chinese patients revealed that severity of pain and duration of pain had a higher risk to anxiety symptoms while higher monthly income, better family functioning and higher pain self-efficacy had a lower risk to anxiety symptoms [10]. Unfortunately, the tool used was Generalized Anxiety Disorder-7 (GAD-7) scale. Being a self-report tool, it could potentially be subjected to recall bias and the authors conspicuously acknowledged this in their limitation.

Bair et al. [11] pursued to assess depression and anxiety in musculoskeletal pain patients attending primary care clinic in Indianapolis, United States of America. They concluded that failure to recognize and treat comorbid GAD in patients with chronic pain can lead to inadequate pain management, increased disability, and poorer overall health outcomes. Consequently, improvements in anxiety predict better pain outcomes and should be addressed to optimize the effectiveness of pain-specific therapies.

Therefore, this study aimed to determine the prevalence and factors associated with GAD among patients with chronic pain in a hospital setting using a diagnostic, structured interview instrument.

## Methods

### Study design and setting

This was a cross-sectional study using universal sampling method conducted among chronic pain patients attending an outpatient setting of a pain clinic at the Raja Perempuan Zainab II Hospital (HRPZ II), Kelantan, Malaysia. 01 February 2021 until 31 May 2021 were the recruitment period for this study. Hospital Raja Perempuan Zainab II (HKK) is a government hospital located in Kota Bahru, Kelantan, Malaysia. It is the biggest hospital in the state Kelantan with 920 beds. Privacy and confidentiality were ensured by safe data-keeping accessible only by researchers during analysis and presentation of data post-analysis.

Sample size was established by using the formula for single proportion which is [DEFF*Np $(1-p)$]/ [(d2/Z21-$\alpha$/2*(N-1)+p*(1-p)]. By using the prevalence of anxiety in patients with chronic pain attending the rheumatology clinic [12], a sample size of 201 patients was determined, including a 20% dropout rate. The sample size was deemed adequate as the power of study from this calculation was 80% with 5% desired precision at 95% confidence interval.

Inclusion criteria were as follows: consecutive adult patients aged 18 years or older who sustained chronic pain attending treatment at the pain clinic in HRPZ II; patients who were able to comprehend Malay or English languages and had consented were recruited. Chronic pain is defined as pain that persists or recurs for more than 3 months according to the International Association for the Study of Pain (IASP) [1].

Patients who had impaired cognitive functions such as having severe psychosis, psychomotor agitation/ retardation or mental disability and those admitted to the ward were excluded.

### Measures

A self-administered questionnaire was used to obtain data on sociodemographic which were age, gender, marital status, level of education, employment status, monthly household income, and ethnicity. Whereas clinical data on diagnosis, duration of the pain and type of analgesic used were derived from the participant's medical records.

Monthly household income was subcategorized as B40, M40 and T20 groups with a monthly income of less than RM4850 (USD1197.98), RM4850—RM10 959 (USD2706.93), and above RM10 959, respectively [13].

### Hospital Anxiety and Depression Scale (HADS)

Anxiety symptoms were screened using HADS. HADS was initially developed by Zigmond and Snaith [14] to assess the symptoms of anxiety and depression. It consisted of 14 items, of which seven of them were related to anxiety (HADS-A) while the remaining items were related to depression (HADS-D). Each item was rated from a scale of 0 to 3, which means that the minimum and maximum scores for both subscales were 0 and 21, respectively. The HADS had been translated to the Malay language and validated [15] with good internal consistency for both subscales (Cronbach's $\alpha$ of 0.88 and 0.79 for anxiety and depression subscales, respectively) [16]. The best cut-off point was between 8 to 9. The scores could be further categorized depending on severity into normal (0–7), mild (8–10), moderate (11–14), and severe (15–21) [17].

### Mini—International Neuropsychiatric Interview, English and Malay versions (M.I.N.I)

Those participants who scored 8 and above on the HADS were then interviewed using M.I.N.I by the same researcher. M.I.N.I was designed as a short diagnostic tool to affirm a variety of

psychiatric disorders including GAD [18]. The inter-rater reliability of M.I.N.I was satisfactory (0.67–0.85), while the Malay version was acceptable as well as clinically relevant in making a diagnosis of GAD [19].

### Numerical Rating Scale (NRS)

The pain intensity was assessed using NRS. NRS was a one-dimensional measure of pain intensity in adults including those with chronic pain. NRS was a segmented numeric version of the Visual Analogue Score in which a participant selects a whole number (0 to 10 integers) that best reflects the intensity of the pain. The corresponding pain levels were as follows: 0 no pain, 1–3 mild pain, 4–6 moderate pain, 7–10 severe pain [20].

### Data analysis

Data was analyzed using Statistical Package for Social Science (SPSS) version 26.0. Descriptive statistics were used to analyze the sociodemographic characteristics of study participants. The continuous variables were reported as mean and standard deviation. Bivariate analysis using Pearson Chi-Square's tests, independent T test and Fisher Exact test were done to analyze significant association between dependent variables and independent variables. Significant associations were further analyzed with simple logistic regression (SLogR). Subsequently, multivariate analysis was performed to determine the best contributing factor from independent variables towards the outcome variables. The independent variables with a significant value of $p < 0.05$ from univariate analysis were included in the multivariate analysis using multiple logistic regression.

### Ethical consideration

Ethical approval was granted from the research ethics committee, Universiti Teknologi MARA (approval code: REC/03/2020(MR/55) and the National Medical Research and Ethics Committee (MREC) of the Ministry of Health (MOH), Malaysia via the National Medical Research Registry (NMRR) (Protocol no NMRR-19-4213-52458). In addition, approvals were obtained from Clinical Research Centre as well as Head of Pain Clinic from Hospital Raja Perempuan Zainab II Kota Bharu. Both verbal and written consent were obtained from the participants.

## Results

### Sociodemographic characteristics of study participants

A total of 201 participants were recruited in this study. Table 1 depicts the socio-demographic details of the study participants. More than half of the participants were male (n = 111, 55.2%) while the remaining were female (n = 90, 44.8%) with a mean age 45.55 (12.38) years. Variable for age was normally distributed with Kolmogorov-Smirnov value of 0.2. The majority of the participants were Malays (n = 196, 97.5%), married (n = 147, 73.1%) and self-employed (n = 65, 32.3%). For monthly income, more than three-quarter of the participants were in the Bottom 40 group (B40) (n = 162, 80.6%) and the remaining were in Middle 40 (M40) group (n = 32, 15.9%) as well Top 20 (T20) group (n = 7, 3.5%).

### Clinical characteristic of the participants

From Table 2, most of the participants were diagnosed to have chronic musculoskeletal pain (n = 114, 56.7%) with most of them enduring mild pain (n = 109, 54.2%). About two thirds of the participants used non-opioid analgesics (n = 141, 70.1%) while 3.5% of the participants

**Table 1. Sociodemographic of the participants (n = 201).**

| Variables | Mean (SD) | n (%) |
|---|---|---|
| **Age (year)** | 45.55(12.38) | |
| **Gender** | | |
| Male | | 111(55.2) |
| Female | | 90(44.8) |
| **Ethnicity** | | |
| Malay | | 196(97.5) |
| Non-Malay | | 5(2.5) |
| **Religion** | | |
| Islam | | 197(98.0) |
| Buddhist | | 4(2.0) |
| **Marital status** | | |
| Single | | 32(15.9) |
| Married | | 147(73.1) |
| Widow | | 14(7.0) |
| Divorced | | 8(4.0) |
| **Education level** | | |
| No formal education | | 14(7.0) |
| Primary | | 25(12.4) |
| Secondary | | 83(41.3) |
| Tertiary | | 79(39.3) |
| **Employment status** | | |
| Government sector | | 50(24.9) |
| Private sector | | 34(16.9) |
| Self-employed | | 65(32.3) |
| Unemployed | | 40(19.9) |
| Others (Pensioner) | | 12(6.0) |
| **Monthly income** | | |
| B40 <RM4850.00 | | 162(80.6) |
| M40 RM4851.00 –RM10970.00 | | 32(15.9) |
| T20 >RM10,971.00 | | 7(3.5) |

were prescribed opioid analgesics (n = 7, 3.5%). The mean duration of pain was 4.88 (4.82) years.

## Prevalence of generalized anxiety disorder and its associated factors

By means of M.I.N.I, 38 participants were diagnosed with GAD (18.9%).

For univariate analysis, Pearson Chi-Square and Fisher Exact Test were conducted. Gender (p<0.001), ethnicity (p = 0.005), and pain severity (p<0.001) were found to be statistically significant (refer Table 3).

Further analysis using multiple logistic regressions (refer Table 4) with forward conditional was used to test for independent association. Results revealed that three independent variables were significantly associated with GAD. Participants with an increase of 1-year duration of pain had 1.30 times the odds of having GAD (95% CI 1.03, 1.63, p = 0.028) when adjusted for gender and pain severity. Female participants had 7.94 times the odds compared to male participants of having GAD (95% CI 2.34, 26.93, p = 0.001). Participants with moderate to severe pain had 18.75 times the odds of having GAD (95% CI 1.23, 285.13, p = 0.035) when adjusted for duration of pain and gender (refer to Table 3).

**Table 2. Clinical characteristics of the participants.**

| Variables | Mean (SD) | n (%) |
|---|---|---|
| **Diagnosis** | | |
| Chronic primary pain | | 3(1.5) |
| Chronic cancer pain | | 6(3.0) |
| Chronic post-surgical/post traumatic pain | | 34(16.9) |
| Chronic neuropathic pain | | 23(11.4) |
| Chronic headache/orofacial pain | | 20(10.0) |
| Chronic visceral pain | | 1(0.5) |
| Chronic musculoskeletal pain | | 114(56.7) |
| **Pain severity** | | |
| No pain | | 1(0.5) |
| Mild pain | | 109(54.2) |
| Moderate pain | | 82(40.8) |
| Severe pain | | 9(4.5) |
| **Type of analgesics** | | |
| Non opioid | | 141(70.1) |
| Opioid | | 7(3.5) |
| Mixed | | 53(26.4) |
| **Duration of pain (years)** | 4.88(4.82) | |

## Discussion

Our hospital-based study had showed that the prevalence of generalized anxiety disorder (GAD) among chronic pain patients was 18.9%. This conclusion was consistent with previous studies conducted both in a Canadian and Indian populations which had recorded the GAD prevalence among chronic pain participants as 17.4% [21] and 18% [4], respectively.

In terms of gender profiling, the prevalence of GAD in pain patients was significantly higher among female than male participants. In fact, females had a 7.94 times higher chance of having GAD. This outcome concurred with previous research performed in the U.S general population, which showed that females were affected twice as often as males [22]. Moreover, based on National Comorbidity Survey (NCS), the prevalence of generalized anxiety disorder among females and male participants was 3.4% and 1.9%, respectively [23]. It was apparent that females were more likely than male participants to disclose their anxiety symptoms and consequently sought treatment for it.

For pain severity, this survey had demonstrated a significant association between pain severity and GAD among chronic pain participants. Participants having high pain scores were more likely to develop GAD. This finding was compatible with a population base study investigating differences in pain severity for arthritis, migraine, and back pain in which high pain scores were reported among patients with GAD compared to patients with pain alone [18]. Moreover, Jun et al. [24] established that high pain scores as well as increased level of anxiety were linked to elevated pain catastrophizing.

This study revealed a significant association between pain duration and GAD in chronic pain participants. This inference was in accordance with a study conducted among a German population, among GAD participants who had more likely developed the condition after enduring chronic pain for at least one year [25].

The insignificant association between marital status, educational level, household income and GAD among patients with chronic pain suggested that family responsibilities, poor education level or stress due to low income were not related to anxiety in our population.

**Table 3. Association between independent variables with GAD.**

| Variables | Group n(%) (n = 201) | | P values |
|---|---|---|---|
| | GAD No (n = 163) | GAD Yes (n = 38) | |
| **Age** | 45.49(12.30) | 45.79(12.90) | 0.894[a] |
| **Gender** | | | <0.001[b] |
| Male | 100(61.3) | 11(28.9) | |
| Female | 63(38.7) | 27(71.1) | |
| **Ethnicity** | | | 0.005[b] |
| Malay | 162(99.4) | 34(89.5) | |
| Chinese | 1(0.6) | 4(10.5) | |
| Indian | 0(0.0) | 0(0.0) | |
| Others | 0(0.0) | 0(0.0) | |
| **Marital Status** | | | 0.856[c] |
| Single | 27(16.6) | 5(13.2) | |
| Married | 119(73.0) | 28(73.7) | |
| Widow | 11(6.7) | 3(7.9) | |
| Divorced | 6(3.7) | 2(5.3) | |
| **Level of education** | | | 0.189[b] |
| No formal education | 14(8.6) | 0(0.0) | |
| Primary | 21(12.9) | 4(10.5) | |
| Secondary | 68(41.7) | 15(39.5) | |
| Tertiary | 60(36.8) | 19(50.0) | |
| **Employment status** | | | 0.050[a] |
| Government sector | 38(23.3) | 12(31.6) | |
| Private sector | 27(16.6) | 7(18.4) | |
| Self employed | 60(36.8) | 5(13.2) | |
| Unemployed | 28(17.2) | 12(31.6) | |
| Others (Pensioner) | 10(6.1) | 2(5.3) | |
| **Monthly income** | | | 0.950[a] |
| <RM4850.00 | 131(80.4) | 31(81.6) | |
| RM4851-RM10970 | 26(16.0) | 6(15.8) | |
| >RM10971 | 6(3.7) | 1(2.6) | |
| **Diagnosis of illness** | | | 0.750[a] |
| Chronic primary pain | 2(1.2) | 1(2.6) | |
| Chronic cancer pain | 4(2.5) | 2(5.3) | |
| Chronic post-surgical | 29(17.8) | 5(13.2) | |
| Chronic neuropathic pain | 20(12.3) | 3(7.9) | |
| Chronic headache/ orofacial pain | 16(9.8) | 4(10.5) | |
| Chronic visceral pain | 1(0.6) | 0(0.0) | |
| Chronic musculoskeletal pain | 91(55.8) | 23(60.5) | |
| **Pain severity** | | | <0.001[a] |
| No pain | 1(0.6) | 0(0.0) | |
| Mild pain | 103(63.2) | 6(15.8) | |
| Moderate pain | 53(32.5) | 29(76.3) | |
| Severe pain | 6(3.7) | 3(7.9) | |
| **Types of analgesics** | | | 0.148[b] |
| Non opioid | 113(69.3) | 28(73.7) | |
| Opioid | 4(2.5) | 3(7.9) | |
| Mixed | 46(28.2) | 7(18.4) | |
| **Duration of pain** | 4.95(4.79) | 4.55(4.98) | 0.647[c] |

**Table 4. Results for variables associated with GAD among participants by multiple logistic regression.**

| Variable | Multiple Logistic Regression | |
|---|---|---|
| | Adjusted OR (95%CI) | P value |
| **Duration of pain** | 1.30(1.03,1.63) | 0.028 |
| **Gender** | | 0.001 |
| Male | 1 | |
| Female | 7.94(2.34, 26.93) | |
| **Pain severity** | | 0.035 |
| No pain to mild | 1 | |
| Moderate to severe | 18.75(1.23,285.13) | |

[a] Forward LR Multiple Logistic Regression model was applied.

Multicollinearity and interaction term were checked and not found.

Hosmer-Lemeshow test, (p = 0.944), classification table (overall correctly classified percentage = 91.0%) and area under the ROC curve (96.4%) were applied to check model fitness.

From the results of this study, there are some recommendations that could be proposed. For policy makers, improvement in current standardized care protocols that emphasize anxiety-reduction techniques in pain management could be investigated. For clinicians, training can be provided regarding the latest techniques to help patients with GAD, especially those with moderate to severe pain categories. For example, usage of Virtual Reality technology for relaxation and distraction during painful procedures. Additionally, recommendations for public health practitioners include community engagement by conducting public health campaigns to improve awareness about chronic pain, the importance of managing anxiety, and the availability of reliable resources for patients to get information. Pertaining to future study, it can include research into technological interventions such as telehealth services to reduce anxiety via remote consultations or non-pharmacological therapies such as hydrotherapy in pain clinic settings.

This study did have limitations. Firstly, being a cross-sectional survey, it did not allow for cause and effect associations to be examined. Secondly, the study outcomes could not represent the whole chronic pain population in Kelantan or Malaysia because this study was conducted only in a single center. Thirdly, other associated factors for GAD such as family history, social support, ongoing physical and medical illness were not investigated which could have impacted the study outcome.

## Conclusions

There was a high prevalence of generalized anxiety disorder among chronic pain patients. Factors significantly associated with GAD among chronic pain patients were gender, duration of pain and severity of pain.

## Acknowledgments

The authors would like to express heartfelt gratitude to the Department of Psychiatry, Faculty of Medicine, Universiti Teknologi MARA (UiTM) for their assistance and support.

## Author Contributions

**Conceptualization:** Norlaila Abd Rahman, Suthahar Ariaratnam, Nurul Azreen Hashim.

**Data curation:** Norlaila Abd Rahman, Zahir Izuan Azhar.

**Formal analysis:** Zahir Izuan Azhar.

**Methodology:** Norlaila Abd Rahman, Suthahar Ariaratnam, Nurul Azreen Hashim, Zahir Izuan Azhar.

**Software:** Zahir Izuan Azhar.

**Supervision:** Suthahar Ariaratnam, Nurul Azreen Hashim, Zahir Izuan Azhar.

**Validation:** Zahir Izuan Azhar.

**Visualization:** Zahir Izuan Azhar.

**Writing – original draft:** Norlaila Abd Rahman, Suthahar Ariaratnam, Nurul Azreen Hashim, Zahir Izuan Azhar.

**Writing – review & editing:** Norlaila Abd Rahman, Suthahar Ariaratnam, Nurul Azreen Hashim, Zahir Izuan Azhar.

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
