## [Decision Letter · Decision Letter 0]

14 May 2024

PONE-D-24-04624Generalized anxiety disorder among patients with chronic pain in MalaysiaPLOS ONE

Dear Dr. Ariaratnam,

Thank you for submitting your manuscript to PLOS ONE. After careful consideration, we feel that it has merit but does not fully meet PLOS ONE’s publication criteria as it currently stands. Therefore, we invite you to submit a revised version of the manuscript that addresses the points raised during the review process.

We look forward to receiving your revised manuscript.

Kind regards,

Syed Sharizman Syed Abdul Rahim, MBBch BAO, MPH, DrPH

Academic Editor

PLOS ONE

Journal Requirements:

2. In the online submission form, you indicated that data is avaiable upon request. The data underlying the results presented in the study are available from Dr Norlaila Binti Abd Rahman at +6014-8082895

Additional Editor Comments:

Kindly do the needful based on the Reviewers' comments

Reviewers' comments:

Reviewer's Responses to Questions

**Comments to the Author**

1. Is the manuscript technically sound, and do the data support the conclusions?

Reviewer #1: Partly

Reviewer #2: Yes

2. Has the statistical analysis been performed appropriately and rigorously? 

Reviewer #1: Yes

Reviewer #2: Yes

3. Have the authors made all data underlying the findings in their manuscript fully available?

Reviewer #1: No

Reviewer #2: No

4. Is the manuscript presented in an intelligible fashion and written in standard English?

Reviewer #1: Yes

Reviewer #2: Yes

5. Review Comments to the Author

Reviewer #1: 1. Title: the title might be misleading. It should be the “prevalence and factors associated with Generalized anxiety disorder among patients with chronic pain: a single center cross sectional study in Malaysia”

2. Abstract

a. Introduction: OK

b. Method: better to mention what type of analysis used to determine factors associated with GAD

c. The phrases “Multiple logistic regression showed the existence of chronic pain amongst GAD patients,…” this phrase is quite confusing. This study population is among the chronic pain patients or GAD patients? no need to mention this as it is confusing. Just mention what are the associated factors associate with GAD

d. Conclusion: “Chronic pain was a prevalent condition in GAD patients”. I think the more accurate phrase is “GAD is a prevalent condition among chronic pain patients”.

3. Introduction

a. There is no problem statement mentioned in the introduction. Please state what are the challenges or issues that need to be addressed in regards to GAD in chronic pain patients. in addition, the author needs to mention why it is important to do this study and its significance.

b. There is no statement regarding what the current knowledge on the factors is associated with GAD among chronic pain patients. I believe there are previous studies that explore this issue. For example, a narrative review by Lokapur et al., (2023) revealed that a total of 84 articles were included in the analysis of depressive and/or anxiety disorders with chronic pain conditions in the Indian populations (DOI: 10.4103/ijpn.ijpn_26_21)

c. If there is various studies that are explored regarding factor associated with GAD in chronic pain patients, what are the new knowledge that could be obtained from this study?

4. Method

a. Regarding study population, is there any reason on why only one hospital is selected for the study population? This is because in the title the author specifically mentioned “ Generalized anxiety disorder among patients with chronic pain in Malaysia” whereby I believe there is more than one hospital with pain clinic in Malaysia.

b. Sample size calculations, please mention what are the formula used (rather than mentioning about the tool used, please be specific), the prevalence of GAD among patients in rheumatology clinic, what are the confidence intervals and desired precision. I believe the sample size the author acquire is too low and not adequate to detect the desired prevalence of GAD in the study population.

c. Please describe how the study population are diagnosed with chronic pain. For how long did they suffer from the pain before they are diagnosed? How about pain clinic patients that were admitted? Do they included in this study?

d. Regarding study location, please describe about it, the population demography etc.

5. Results

a. Regarding descriptive analysis, please state the distribution of continuous variable (age) whether it is normal distributed or not

b. The statement “The prevalence of GAD: By means of M.I.N.I, 38 participants were diagnosed with GAD (18.9%). Hence, the prevalence of GAD in our population was 18.9%”, (it is better to avoid repetition)

c. It is interesting if the author included the univariate analysis results in the article. Authors can include the univariate analysis separately or in the same table as the multivariate analysis table.

d. Similarly, it is interesting if the author can include in the multivariate analysis table, the comparison of have GAD/no GAD for all the variables analyzed. This information is valuable for the readers to make sense of the data that the authors concluded.

e. Univariate analysis also is important to look for which variables that the authors included in the multivariate analysis. As far as my understanding, the authors only included the three variables in the multivariate analysis which are duration of pain, gender, and pain severity. The authors might miss other important variables to include in the multivariate analysis, such as education level, income status, and employment status, due to the use of p<0.05 as cut off points. According to Hesmer and Lemeshaw, the use of traditional levels such as 0.05 often fails to identify variables known to be important (Bursac et al. 2008).

6. Discussion

a. I believe that the study population is likely to be lower than is should be. By evidence of wide confidence interval for the association of gender (AOR:7.94; 95% CI:2.34, 26.93) and pain severity (AOR:18.75; CI:1.23,285.13) with GAD. Avijit Hazra (2017) stated that a wide confidence interval (CI) can indicate that the sample size is small thus producing less precise results. Please add in the limitation on why the study population is lower (if the author think otherwise/sample size is actually adequate, please justify it with reference)

b. For every significant and non-significant finding, please state what are the significance and please provide recommendations for policy maker, clinicians, public health practitioner, and future study.

c. I believe there is a lot to discuss about the study findings. Please discuss on why the other important variable (marital, education, employment, income, diagnosis, and type of analgesic) is not significantly associated with GAD.

7. Conclusion

a. In the conclusion, it is better to conclude the findings according to your study objectives.

8. References

a. Some references in the text are not consistent. The is a mix of usage of Vancouver and APA reference style. For example, in the discussion section, third paragraphs, “This finding was compatible with a population base study investigating differences in pain severity for arthritis, migraine, and back pain in which high pain scores were reported among patients with GAD compared to patients with pain alone (Csupak et al., 2018). Moreover, Jun et al. [21] established that high pain scores as well as increased level of anxiety were linked to elevated pain catastrophizing.” Please refer to the journal’s standard.

b. please refer journal's standard

Reviewer #2: Dear Authors,

Thank you for submitting your manuscript to the journal. Your study on the prevalence of Generalized Anxiety Disorder (GAD) among patients with chronic pain in Malaysia provides valuable insights into a critically under-researched area. The study design, methodology, and statistical analyses appear robust and well-executed. The ethical considerations are appropriately addressed, and the conclusions drawn from the analyses are supported by the data. However, I would like to raise a concern regarding the availability of the data supporting your findings. Ensuring that data are freely available enhances transparency, allows for the verification of results, and supports the reproducibility of the study.In conclusion, with adjustments to the data availability statements and minor enhancements in the reporting of your methods and results, your manuscript would be a significant contribution to the literature. I look forward to your revisions and potentially the broader dissemination of your findings.

Best regards

6. PLOS authors have the option to publish the peer review history of their article (what does this mean?). If published, this will include your full peer review and any attached files.

Reviewer #1: **Yes: **MOHD FAZELI SAZALI

Reviewer #2: No

---

## [Author Response · Author response to Decision Letter 0]

18 Jun 2024

Response to Additional Editor Comments:

1. Is the manuscript technically sound, and do the data support the conclusions?

Reviewer #1: Partly

Reviewer #2: Yes Based on our response to reviewer 1 comments for points 2 and 3 below, this statement has been adequately addressed.

2. Has the statistical analysis been performed appropriately and rigorously

Has the statistical analysis been performed appropriately and rigorously?

Reviewer #1: Yes

Reviewer #2: Yes 

Thank you for the input.

3. Have the authors made all data underlying the findings in their manuscript fully available?

Reviewer #1: No

Reviewer #2: No

Our study had involved human research participant data. Hence, there are ethical and legal restrication on sharing of de-identified or anonymized data because these information contain potentially identifying or sensitive patient information.

However,data requests may be sent to the following ethics committees

1. Chairperson,

 Medical Research & Ethics Committee,

 Ministry of Health Malaysia,

 Kompleks Institut Kesihatan Negara,

 Blok A, No 1,Jalan setia Murni U 13/52,

 Seksyen U 13, Bandar Setia Alam, 

 40170 Shah Alam, Selangor,

 Malaysia.

 Tel: +603-33628888

 Email: mrecsec@moh.gov.my

2. Chair,

 Universiti Teknologi MARA Research

 Ethics Committee,

 Aras 3, Bangunan Wawasan,

 40450 Shah Alam, Selangor,

 Malaysia.

 Tel: +603-55442004

 Fax: +603-55442070

4. Is the manuscript presented in an intelligible fashion and written in standard English?

Reviewer #1: Yes

Reviewer #2: Yes 

Thank you very much

5. Please amend either the abstract on the online submission form (via Edit Submission) or the abstract in the manuscript so that they are identical. Thank you for this statement. We have duly amended this during the submission process.

We have accordingly amended the above so that both abstracts are identical.

Response to reviewer 1’s comments:

1. Title: the title might be misleading. It should be the “prevalence and factors associated with Generalized anxiety disorder among patients with chronic pain: a single center cross sectional study in Malaysia” 

Thank you and we concur with the statement. Hence, the title has been changed in the revised manuscript

1. Abstract

a. Introduction: OK

This is much appreciated.

b. Method: better to mention what type of analysis used to determine factors associated with GAD

Multiple logistic regression analysis was used to determine factors associated with GAD. We have incorporated this statement in the revised manuscript (Abstract: Methods, Line 5-6)

c. The phrases “Multiple logistic regression showed the existence of chronic pain amongst GAD patients,…” this phrase is quite confusing. This study population is among the chronic pain patients or GAD patients? no need to mention this as it is confusing. Just mention what are the associated factors associate with GAD

We agree with the comment. Thus, the relevant phrase has been omitted and rewritten as “Gender (AOR:7.94; 95% CI:2.34, 26.93), duration of the pain (AOR:1.30; 95% CI:1.03,1.63) and pain severity (AOR:18.75; CI:1.23,285.13) were significant factors associated with GAD”. (refer line 3-5)

d. Conclusion: “Chronic pain was a prevalent condition in GAD patients”. I think the more accurate phrase is “GAD is a prevalent condition among chronic pain patients”. 

Thank you for the note. We have rectified appropriately in the revised manuscript (refer line 2)

2. Introduction

a. There is no problem statement mentioned in the introduction. Please state what are the challenges or issues that need to be addressed in regards to GAD in chronic pain patients. in addition, the author needs to mention why it is important to do this study and its significance.

Thank you for pointing this out. We have revised the manuscript to include problem statement, challenges and issues. We believe that this has been have been adequately addressed (Kindly refer to Introduction: para 6-8)

b. There is no statement regarding what the current knowledge on the factors is associated with GAD among chronic pain patients. I believe there are previous studies that explore this issue. For example, a narrative review by Lokapur et al., (2023) revealed that a total of 84 articles were included in the analysis of depressive and/or anxiety disorders with chronic pain conditions in the Indian populations (DOI: 10.4103/ijpn.ijpn_26_21)

We have cited Lokapur et al., (2023) in the manuscript and addressed its weakness. (please refer to Introduction: Para 6)

c. If there is various studies that are explored regarding factor associated with GAD in chronic pain patients, what are the new knowledge that could be obtained from this study?

We have reinforced our aim, mentioned as “Therefore, this study aimed to determine the prevalence and factors associated with GAD among patients with chronic pain in a hospital setting using a diagnostic, structured interview instrument. “(kindly refer to Introduction: last para)

3. Method

a. Regarding study population, is there any reason on why only one hospital is selected for the study population? This is because in the title the author specifically mentioned “ Generalized anxiety disorder among patients with chronic pain in Malaysia” whereby I believe there is more than one hospital with pain clinic in Malaysia. 

We agree with this statement. However, due to limited time and resources as well financial constraints, the study was limited to one center.

b. Sample size calculations, please mention what are the formula used (rather than mentioning about the tool used, please be specific), the prevalence of GAD among patients in rheumatology clinic, what are the confidence intervals and desired precision. I believe the sample size the author acquire is too low and not adequate to detect the desired prevalence of GAD in the study population. 

Thank you for that response. The sample size formula was added using formula for single proportion. Desired precision with confidence interval were stated as well.

The sample size is adequate as the power of the study from sample size calculation is 80%.

(Please refer to Methods: para 2. The original para 2 has been omitted to improve clarity)

c. Please describe how the study population are diagnosed with chronic pain. For how long did they suffer from the pain before they are diagnosed? How about pain clinic patients that were admitted? Do they included in this study?

Thank you for this comment. Only outpatient participants with chronic pain were recruited. Chronic pain is defined as pain that persists or recurs for more than 3 months according to the International Association for the Study of Pain.

Those admitted were not included in the study population.

We have incorporated the above in the revised manuscript (please refer Methods: para 3 line 3-5 & para 4 last line)

d. Regarding study location, please describe about it, the population demography etc. 

Hospital Raja Perempuan Zainab II (HKK) is a government hospital located in Kota Bahru, Kelantan, Malaysia. It is the biggest hospital in Kelantan with 920 beds. 

This statement has been included in the revised manuscript (Kindly refer to Methods: para 1 Line 4-6)

4. Results

a. Regarding descriptive analysis, please state the distribution of continuous variable (age) whether it is normal distributed or not 

Thank you for your input. 

Variable for age is normally distributed with Kolmogorov-Smirnov value of 0.2. (kindly refer to results: para 1 line 3-4)

b. The statement “The prevalence of GAD: By means of M.I.N.I, 38 participants were diagnosed with GAD (18.9%). Hence, the prevalence of GAD in our population was 18.9%”, (it is better to avoid repetition) 

Thank you for pointing this out. We have removed the second statement and maintain “By means of M.I.N.I, 38 participants were diagnosed with GAD (18.9%). "Hence, the prevalence of GAD in our population was 18.9%” has been removed

c. It is interesting if the author included the univariate analysis results in the article. Authors can include the univariate analysis separately or in the same table as the multivariate analysis table. 

Thank you very much. We have added Table 3 to display the results for univariate results and a para to describe it in the text (refer to Table 3 & para 2 under sub section of Prevalence of Generalized Anxiety Disorder and its associated factors)

d. Similarly, it is interesting if the author can include in the multivariate analysis table, the comparison of have GAD/no GAD for all the variables analyzed. This information is valuable for the readers to make sense of the data that the authors concluded. 

Table 3 has been added to display the comparison of having GAD and not having GAD for all the variables analyzed. Kindly refer back to point c above.

e. Univariate analysis also is important to look for which variables that the authors included in the multivariate analysis. As far as my understanding, the authors only included the three variables in the multivariate analysis which are duration of pain, gender, and pain severity. The authors might miss other important variables to include in the multivariate analysis, such as education level, income status, and employment status, due to the use of p<0.05 as cut off points. According to Hesmer and Lemeshaw, the use of traditional levels such as 0.05 often fails to identify variables known to be important (Bursac et al. 2008). 

For the univariate analysis, the significant variables: Gender and Pain Severity are included in the multivariate analysis. Besides these two variables, duration of pain and ethnicity are also included. Even though the p-value of variable duration of pain was >0.05, it was also included in multivariate analysis because this is an important variable related to this study.

5. Discussion

a. I believe that the study population is likely to be lower than is should be. By evidence of wide confidence interval for the association of gender (AOR:7.94; 95% CI:2.34, 26.93) and pain severity (AOR:18.75; CI:1.23,285.13) with GAD. Avijit Hazra (2017) stated that a wide confidence interval (CI) can indicate that the sample size is small thus producing less precise results. Please add in the limitation on why the study population is lower (if the author think otherwise/sample size is actually adequate, please justify it with reference) 

Thank you for the input. 

Sample size was established by using the formula for single proportion which is [DEFF*Np(1-p)]/ [(d2/Z21-α/2*(N-1)+p*(1-p)]. By using the prevalence of anxiety in patients with chronic pain attending the rheumatology clinic [Sulaiman et al, 2017], a sample size of 201 patients was determined, including a 20% dropout rate. The sample size was deemed adequate as the power of study from this calculation was 80% with 5% desired precision at 95% confidence interval. (Kindly refer to Method: point b. above)

b. For every significant and non-significant finding, please state what are the significance and please provide recommendations for policy maker, clinicians, public health practitioner, and future study. 

Thank you for your comments. These points have been addressed accordingly (Please refer discussion: para 5 and 6).

c. I believe there is a lot to discuss about the study findings. Please discuss on why the other important variable (marital, education, employment, income, diagnosis, and type of analgesic) is not significantly associated with GAD. 

Thank you very much for this pertinent response.

Since, this study was reasonably conducted and hence, we can merely mention that it was more likely that there was no difference in the other important variable mentioned (i.e. marital, education, employment, income, diagnosis, and type of analgesic). However, if there were likely weaknesses (some have already been quoted under limitation) that could have prevented our study from finding a potential difference.

We hopefully believe that the correction we have made will meet with your kind approval. 

6 Conclusion

a. In the conclusion, it is better to conclude the findings according to your study objectives. 

We agree with the statement. The conclusion has been revised according to the study objectives. (Please refer to conclusion: para 1. Thus, previous para 1 and 2 have been deleted)

7. References

a. Some references in the text are not consistent. The is a mix of usage of Vancouver and APA reference style. For example, in the discussion section, third paragraphs, “This finding was compatible with a population base study investigating differences in pain severity for arthritis, migraine, and back pain in which high pain scores were reported among patients with GAD compared to patients with pain alone (Csupak et al., 2018). Moreover, Jun et al. [21] established that high pain scores as well as increased level of anxiety were linked to elevated pain catastrophizing.” Please refer to the journal’s standard.

b. please refer journal's standard 

Thank you for input. We have modified the referencing style according to PLOS ONE guidelines.

Response to reviewer 2’s comments:

Thank you very much for the comment. 

We hopefully believe that the correction we have made will meet with your kind approval.

---

## [Editor Report · Decision Letter 1]

16 Jul 2024

Prevalence and factors associated with Generalized anxiety disorder among patients with chronic pain: A single center cross sectional study in Malaysia

PONE-D-24-04624R1

Dear Dr. Ariaratnam,

We’re pleased to inform you that your manuscript has been judged scientifically suitable for publication and will be formally accepted for publication once it meets all outstanding technical requirements.

Kind regards,

Syed Sharizman Syed Abdul Rahim, MBBch BAO, MPH, DrPH

Academic Editor

PLOS ONE
---

## [Editor Report · Acceptance letter]

19 Jul 2024

PONE-D-24-04624R1 

PLOS ONE

Dear Dr. Ariaratnam, 

I'm pleased to inform you that your manuscript has been deemed suitable for publication in PLOS ONE. Congratulations! Your manuscript is now being handed over to our production team.

Kind regards, 

on behalf of

Associate Professor Syed Sharizman Syed Abdul Rahim 

Academic Editor

PLOS ONE